# Comprehensive Elucidation of the Role of L and 2A Security Proteins on Cell Death during EMCV Infection

**DOI:** 10.3390/v16020280

**Published:** 2024-02-11

**Authors:** Yury Ivin, Anna Butusova, Ekaterina Gladneva, Anatoly Gmyl, Aydar Ishmukhametov

**Affiliations:** 1FSASI “M.P. Chumakov Federal Scientific Center for Research and Development of Immunobiological Drugs of the Russian Academy of Sciences (Polio Institute)”, 118819 Moscow, Russia; amadreaera@gmail.com (A.B.); gladneva_ee@chumakovs.su (E.G.); ishmukhametov_aa@chumakovs.su (A.I.); 2Institute of Translational Medicine and Biotechnology, First Moscow State Medical University (Sechenov University), 119991 Moscow, Russia

**Keywords:** EMCV, leader protein, 2A protein, security protein, apoptosis, viral infection, nuclear-cytoplasmic traffic, cell death

## Abstract

The EMCV L and 2A proteins are virulence factors that counteract host cell defense mechanisms. Both L and 2A exhibit antiapoptotic properties, but the available data were obtained in different cell lines and under incomparable conditions. This study is aimed at checking the role of these proteins in the choice of cell death type in three different cell lines using three mutants of EMCV lacking functional L, 2A, and both proteins together. We have found that both L and 2A are non-essential for viral replication in HeLa, BHK, and RD cell lines, as evidenced by the viability of the virus in the absence of both functional proteins. L-deficient infection led to the apoptotic death of HeLa and RD cells, and the necrotic death of BHK cells. 2A-deficient infection induced apoptosis in BHK and RD cells. Infection of HeLa cells with the 2A-deficient mutant was finalized with exclusive caspase-dependent death with membrane permeabilization, morphologically similar to pyroptosis. We also demonstrated that inactivation of both proteins, along with caspase inhibition, delayed cell death progression. The results obtained demonstrate that proteins L and 2A play a critical role in choosing the path of cell death during infection, but the result of their influence depends on the properties of the host cells.

## 1. Introduction

Encephalomyocarditis virus (EMCV) is a small virus with approximately 30 nm non-enveloped virions carrying a single-stranded (+)RNA genome that encodes 13 proteins [1,2]. It belongs to the *Cardiovirus* genus of the *Picornaviridae* family and serves as a model for studying the properties of picornaviruses [3]. EMCV strains were originally isolated from mammals in the mid-20th century: mice, primates, and pigs [4,5,6]. The Mengo strain studied here was isolated from a rhesus monkey in 1948 [7]. It has been shown that EMCV may infect a wide range of wild, farmed, and domestic animals and can cause a range of diseases, such as myocarditis and sudden death in pigs, that may have economic consequences [3]. 

Two of the 13 proteins of EMCV, L and 2A, are known “security” proteins or virulence factors. These proteins modify the intracellular environment and enable the virus to replicate efficiently [8]. Security proteins play a crucial role in the interaction between the host and the pathogen, often leading to the development of pathological processes in infected cells [9]. 

According to the hypothesis of a virus-induced cytopathic effect (CPE), cell death can be seen as a result of a competition between innate immune defenses and antidefense viral tools [9]. Wild-type (WT) EMCV infection, as well as infections by many other picornaviruses, typically leads to a necrotic CPE, characterized by nuclear shrinkage, cell swelling, and plasma membrane rupture [10,11]. Although the detailed molecular mechanism of the picornavirus-driven CPE is still unclear, it is likely connected to an alternative activation pathway(s) of apoptosis [9]. Apoptosis is another type of cell death that can be induced during picornavirus, and particularly, cardiovirus infection [12,13]. It generally occurs during picornavirus infection due to the dysfunction or inhibition of antiapoptotic proteins, which are responsible for shifting the intracellular balance of cell death factors towards necrosis [9]. Apoptotic cell death does not involve membrane permeabilization and aims to prevent the spread of infection within the organism [14]. Typical markers of apoptosis include the degradation of cellular DNA, membrane blebbing, and formation of apoptotic bodies, which are later eliminated by the immune system [15,16]. Apoptosis can be initiated by extrinsic or intrinsic signal pathways triggered by surface death receptors or cytoplasmic sensors, respectively [17]. Activation of both apoptotic pathways leads to the activation of specific cysteine-proteases called caspases. Initiator caspases (caspase-8 and caspase-9 for extrinsic and intrinsic pathway, respectively) are activated at the early stage of the apoptotic cascade and subsequently activate effector caspases (e.g., caspase-3), which result in extensive protein degradation and the aforementioned apoptotic events.

To avoid apoptotic cell death and support viral replication and survival, picornaviruses possess security proteins L and 2A, which play a key role in counteracting the apoptotic death pathway [8]. The leader protein (L) of EMCV is located at the N-terminus of polyprotein and is cleaved from capsid precursor by the 3C/3CD protease [18]. The L protein is a 67 amino acid protein that does not exhibit enzymatic activity and has two functional motifs/domains. The most studied motif is the Zinc-finger domain, which has four residues (1 His and 3 Cys) coordinating Zn^2+^-ions and is responsible for the majority of the known security functions of the protein [19]. Mutations of only two amino acids, Cys19/Ala and Cys22/Ala, were demonstrated to destroy the zinc finger domain and render the L protein inactive [11,20]. 

The L protein of EMCV has been shown to exhibit anti-apoptotic properties in HeLa cells upon viral infection [11]. Specifically, the L protein was demonstrated to activate a hyperphosphorylation of nucleoporins or/and to bind Ran-GTPase, which inhibits active trafficking through the nuclear membrane of the host cell [19,20,21,22]. Investigation of the protein synthesis in the infected cells has revealed a prominent role of the L protein in host translation inhibition [23]. Cardiovirus L protein was shown to suppress such factors as NF-κB and interferon regulatory factor 3, resulting in the inhibition of interferon and antiviral cytokine production [24,25,26].

The 2A protein of EMCV (143 a.a.) is located between structural and non-structural proteins in the polyprotein. It has an NPG-P motif at the C-terminus, called StopGo, through which primary processing between 2A and 2B is carried out [27]. Apart from viral polypeptide processing, the 2A protein is believed to be involved in multiple interactions with cellular pathways. 2A contains a nuclear localization signal (NLS)-like motif that is a homologue to yeast NLS. Not surprisingly, 2A was shown to have nuclear localization [28]. Since proteins L and 2A were shown to interact in co-immunoprecipitation experiments, 2A was suggested to transport L protein to the nucleus and was hypothesized to take part in L functionality [29]. Moreover, similar to L protein, 2A was shown to exhibit anti-apoptotic activity, though a different cell line, BHK-21, was used [30]. Studies examining the ectopic expression of the 2A protein in pig cells (PK15) have also suggested that the 2A protein may have an anti-apoptotic role by potentially interacting with Annexin-A2 [31]. 

Recent studies have revealed a significant role of 2A in the translation regulation of EMCV proteins. 2A was shown to participate in a −1 ribosomal shift as an essential trans-activator [2,32]. The frameshifting was shown to occur at the “slippery” sequence G_GUU_UUX prior to the N-terminus of a 2B-coding region and was shown to require the formation of an RNA-protein complex between a downstream stem-loop and the 2A protein [32]. This mechanism would explain how 2A controls the expression of viral genes. Indeed, the efficiency of frameshifting is positively correlated with 2A concentration, which increases during the infection cycle. Based on 2A structural data, the previously known «NLS-like» motif is precisely involved in the RNA-protein interaction [33]. 

In sum, while both L and 2A are well described as anti-apoptotic tools of cardioviruses, none of the reports investigated their functions simultaneously under the same conditions. 

Previously, we demonstrated that a simultaneous inactivation of the anti-apoptotic EMCV L protein and inhibition of apoptosis abolished CPE during the infection [10]. Under these conditions, the virus continues to replicate without inducing cell death, so we hypothesize that the collision itself between viral and host defense mechanisms is responsible for the pathological processes in infected cells and eventually cell death. Based on this hypothesis, the inactivation of the second safety protein, 2A, may lead to a decrease in pathology development in the context of apoptosis inhibition.

The security proteins of *Picornaviridae* members usually perform similar functions by different mechanisms [8]. For instance, nucleocytoplasmic disorder during the enterovirus infection depends on the activity of 2Apro, which proteolytically degrades nucleoporins [34]. Nevertheless, despite 2Apro activity, it is not essential for viral propagation [35]. Cardioviruses, which lack security proteases, execute this process through the L-mediated phosphorylation of nucleoporins [21]. It is currently not known whether protein EMCV 2A is involved in the process of phosphorylation of nucleoporins.

In this study, we focused on comprehensive research of the properties of the L and 2A proteins in the same context. Using three different cell lines, we investigated the involvement of security proteins in counteracting cell defense in L-deficient, 2A-deficient, and double-deficient viral backgrounds. 

## 2. Materials and Methods

### 2.1. Cells and Viruses 

HeLa-B [12], HeLa-3E [36], BHK-21 and RD cells were cultured in Dulbecco’s modified Eagle’s medium (DMEM, Polio Institute, Russia) supplemented with 10% fetal bovine serum (Gibco, Billings, MT, USA) and 100 µg/mL of kanamycin at 37 °C in 5% CO_2_. 

The wild-type Mengo strain of EMCV serotype 1 (WT) was derived from the plasmid pM16.1 [37]. The Mengo mutant encoding an L protein with substitutions Cys19 → Ala and Cys22 → Ala, which destroy the Zinc finger motif (referred to as Zfmut), was described previously [26] and derived from the pM16.1Zfmut plasmid.

The Mengo mutants Δ2A and Zfmut&Δ2A were obtained following transfection of BHK-21 cells by RNAs transcribed from newly created pM16.1Δ2A and pM16.1Zfmut&Δ2A, respectively. Δ2A encodes a partially deleted (from 11 to 125 aa) 2A protein, while Zfmut&Δ2A has mutations in both the L and 2A.

### 2.2. Construction of pM16.1Δ2A and pM16.1Zfmut&Δ2A

pM16.1Δ2A was constructed based on the pM16.1 using two-step overlapped PCR and cloning method: (1) Two genome fragments were amplified using the Δ2A_1 (5′-CTCCCAAGCAAAGCAGCG-3′)—Δ2A_2 (5′-TATGCAGGATACTTTTCAGATC-3′) and Δ2A_3 (5′-CTGAAAAGTATCCTGCATAAGTTTTAGAGACATCCAAAGGG-3′)—Δ2A_4 (5′-TCAAGACACAACCACTTGCC-3′) primer pairs, respectively; (2) the two fragments with complementary ends were ligated using PCR; and (3) the Δ2A amplicon was cloned into pM16.1 using AflII and SacII restriction enzymes. pM16.1Zfmut&Δ2A was constructed in a similar manner based on the pM16.1Zfmut plasmid [26].

### 2.3. In Vitro Transcription and Transfection 

The plasmids pM16.1, pM16.1Zfmut, pM16.1Δ2A, and pM16.1Zfmut&Δ2A were linearized using the BamHI restriction enzyme and purified using the QIAquick kit from Qiagen. Genomic RNA was then transcribed from these linearized plasmids using T7 RNA-polymerase at 37 °C for 3 h. The RNAs were transfected into BHK cells using the DEAE-dextran transfection kit from Promega. Viral suspensions were obtained from the transfected cells within 1–3 days depending on the viral pathogenicity.

### 2.4. Plaque Titration Assay

For the plaque titration procedure, samples subjected to three cycles of freezing-thawing were used. BHK-21 cells were cultured in 6-well plates until a monolayer was formed. Serial dilutions of the tested viral suspension in DMEM were then distributed onto the cell monolayer (200 µL per well) after removing the growth medium. After 30 min virus adsorption at room temperature (RT) the cells were overlaid with agarose (Sigma, Kawasaki City, Japan, final concentration 0.9%), fetal bovine serum (Gibco, 2%) in Earle’s salt solution supplemented with 3% sodium bicarbonate and kanamycin (1 mg/mL). The plates with cells were then incubated at 37 °C in 5% of CO_2_ for 2–4 days. Cells were fixed using a 5% solution of trichloroacetic acid and stained with a 0.15% crystal violet solution in 25% ethanol after the removing of agarose. Plaques in each dilution of the samples were counted and the titer of the samples represented in plaque-forming units per mL (PFU/mL) were calculated [11]. Then, the titration curves were prepared using GraphPad Prism 10.1.1.

### 2.5. Plaque Cloning Procedure

For the viral cloning aimed to homogenize pools of mutants for subsequent experiments BHK-21 cell line was used. Cells were cultured in 6-well plates until the formation of a monolayer. The viral suspension was diluted to the concentration of 10 PFU/mL in DMEM and then distributed onto the cell monolayer (200 µL per well) after removing the growth medium. After 30 min virus adsorption at room temperature (RT) cells were overlaid with aforementioned agarose solution. After 4 days incubation, the 0.02% neutral-red solution in Earle’s salt solution was added for the plaque visualization. The viral clones were harvested using serological pipettes and used for the subsequent passaging.

### 2.6. Single Cycle Infection Experiments

HeLa, BHK-21 or RD cells were cultured in 12- (for genomic RNA synthesis experiments) or 6-well plates (for TUNEL assay, PI assay and GFP observation) with or without coverslips (Heinz Herenz, Hamburg, Germany) (for viral growth curve and collection of cell lysates). After reaching 50–70% of confluence cells were counted. The growth medium was discarded, cells were washed thrice with DMEM, and a virus was added to provide a multiplicity of infection of 40 PFU/cell. After 30 min of adsorption at room temperature, cells were washed and DMEM was added. After incubation at 37 °C with 5% of CO_2_ for the indicated intervals, cells underwent necessary procedures (RNA isolation, cell fixation prior to fluorescent assays, lysate preparation, or titration). The pan-caspase inhibitor QVD-OPH (QVD, Quinoline-Val-Asp-Difluorophenoxymethylketone, MP Biomedicals, Santa Ana, CA, USA) at a final concentration of 20 µM was used as negative control aimed to inhibit apoptosis in infected cells. As a positive control for inducing apoptosis staurosporine (STS) was used at a final concentration of 500 nM. Viral growth experiments were replicated at least 3 times for all cell lines. Values were expressed as means ± SD (Standard Deviation) of experiments. 2-Way Annova statistics was used to compare variables and calculate *p*-values to identify significant differences. 

### 2.7. TUNEL Assay and Propidium Iodide (PI) Staining

After discarding the medium, mock- and virus-infected cells were fixed. The TUNEL assay was carried out as described previously [10]. For PI staining at 30 min before the desired interval, the growth medium was changed to DMEM containing 10 µg/mL of PI (Sigma). Cells were fixed with 3.7% of paraformaldehyde solution in phosphate-buffered saline (PBS) for 15 min at RT, then washed three times in PBS. Nuclei were stained with Hoechst 33342 (Sigma, 1 µg/mL) for 15 min at RT. Coverslips with cells were placed onto a drop of Mowiol solution (Sigma) on slides.

### 2.8. RNA Extraction and RT-qPCR

Total RNA was isolated from the cell monolayer using a TRI reagent (Sigma) according to the manufacturer’s protocol. 1 µg of RNA was reverse transcribed using Maxima Reverse Transcriptase (ThermoScientific, Waltham, MA, USA) primed with randomized hexanucleotides. The RT-qPCR assays for the viral RNA quantification were performed on an ABI 7500 RT-PCR system (Applied Biosystems, Waltham, MA, USA) by using a commercially available real-time PCR kit (Syntol, Moscow, Russia) according to the supplier’s protocol with the oligonucleotide primers MGVL1 (CGCTAGGAATGCGTAGAACA) and MGVR1 (AGCTCGTCCTTGAGGAATGT) [11]. Oligonucleotide MGVP1 (6FAM-TGGGAAACCGCCACTCTTATCCC-BHQ1) was used as the fluoroprobe. The standard curve was generated by using serial dilutions of a viral genomic RNA isolated from cesium chloride gradient-purified viral particles of WT EMCV. The thermal cycling protocol was as follows: the reaction mixture was first subjected to hot activation at 95 °C for 5 min, followed by 45 cycles of denaturation at 95 °C for 15 s and annealing/extension at 61 °C for 30 s.

### 2.9. Western Blot Analysis

For the Western blot analysis of viral and cellular proteins, cells from 6-well plates were lysed in Laemmli buffer (2% SDS, 50 mM 2-mercaptoethanol, 50 mM Tris-HCl [pH 6.8]) at indicated time points post infection. Lysates were heated for 5 min at 95 °C. Samples were separated on a SDS-PAGE (6–12%) and transferred onto a nitrocellulose membrane in transfer buffer (190 mM glycine, 2.5 mM Tris [pH 8.3]). Membranes were incubated with 5% fat-free milk solution in TBS-T (114 mM NaCl, 17 mM Tris-HCl [pH 8.0], 0.05% Tween 20) for 1 h at RT. Then, membranes were incubated with primary anti-caspase-3 antibodies (Santa Cruz Biotechnology, # sc-7272) in TBS-T milk solution for 1 h. Then, membranes underwent washing with TBS-T, incubation with HRP (horseradish peroxidase)-conjugated secondary antibodies (Promega) for 1 h at RT, and development using the Clarity ECL substrate (Bio-Rad, Hercules, CA, USA). All samples were analyzed with HRP-conjugated antibodies against β-actin (Sigma, St. Louis, MO, USA, #A5316).

## 3. Results

### 3.1. Simultaneous Inactivation of Both L and 2A Security Proteins Does Not Abolish Viral Replication

To establish whether an absence of both functional security proteins L and 2A affects the reproduction of EMCV, we compared the characteristics of the wild-type (WT) virus with those of virus mutants. The first mutant, Zfmut, had mutations in the zinc finger domain of the L protein, which had previously been demonstrated to abolish the security function of the L protein [11]. The second mutant, Δ2A, had a partial deletion of the 2A-coding region. The third mutant, Zfmut&Δ2A, was an EMCV with both a deletion in the 2A-coding region and mutations in L. Previously, we have described that Cys19 → Ala and Cys22 → Ala substitutions in L and partial deletion of the L-coding region (ΔL) lead to the impairment of L functionality identically [10,11]. However, the use of Zfmut rather than ΔL is preferable due to its better reproductive ability [10]. For the deletion of the 2A protein, the region from 11 to 125 amino acids in the viral polypeptide was selected since it includes the known functional motif (amino acids 91–102) responsible for the RNA-loop binding during the activation of −1 frameshifting acts [2,32,33], which was also previously called the nuclear localization signal domain [38].

The WT viral pool was obtained upon transfection of BHK-21 cells with viral RNA, followed by an additional passage in the same cell line. The Zfmut pool was obtained by transfection of the BHK-21 cells, followed by two additional passages. Viruses deficient in 2A protein (Δ2A) and containing both non-functional security proteins (Zfmut&Δ2A) were generated using the same protocol followed by a plaque cloning procedure. All final pools were analyzed by full genome sequencing, and the introduced changes were verified (Figure 1A).

All obtained viruses were able to form plaques in a BHK-21 monolayer, which proved the viability of the mutants. However, the Δ2A-mutants formed smaller plaques than WT and Zfmut (Figure 1B). 

Here, we have demonstrated, for the first time, that EMCV lacking both functional L and 2A security proteins retains the ability to replicate and infect the cells, which proves the facultative role of them in EMCV reproduction. The reproductive ability of the mutants was observed in the next section. 

### 3.2. Functional Inactivation of L and 2A Proteins of EMCV Reduces Viral Yield in HeLa and RD Cell Lines in a Cumulative Manner

Firstly, we aimed to investigate the reproduction ability of all obtained mutants in three different cell lines: HeLa, BHK-21, and RD. Quantification of viral progeny in a single-cycle plaque titration assay revealed that all mutants exhibit a similar ability to produce infectious particles in BHK-21 cells. All tested virus variants, Zfmut, Δ2A, and the “double” mutant Zfmut&Δ2A, showed similar replication kinetics and reached a plateau at 6–8 h post-infection (h.p.i.) with a final viral yield of approximately 100 PFU/cell (Figure 2A). 

Analysis of the reproduction cycle of the mutants in the HeLa cell line demonstrated that functional inactivation of a single security protein in Zfmut and Δ2A mutants leads to a 5-fold decrease in the viral yield at both the exponential (4 h.p.i.) and final (6–8 h.p.i.) reproduction stages (Figure 2B). Simultaneous dysfunction of both L and 2A proteins (Zfmut&Δ2A) leads to an even larger 10-fold reduction in the viral yield throughout the reproductive cycle as compared to WT (Figure 2B). This result was further confirmed in RD cells, in which Zfmut and Δ2A mutants showed a moderate (~5-fold) reduction in viral yields at the final stage of the reproduction cycle (8 h.p.i.), with the double-mutant Zfmut&Δ2A having the lowest reproduction rate (Figure 2C). 

These results suggest that the functional inactivation of the L and 2A security proteins may have a cumulative effect on reducing viral reproduction. It is worth noting that although all the tested virus variants were able to replicate in all tested cell lines, only in BHK-21 cells was the ability of EMCV to reproduce not affected by the inactivation of the security proteins. Thus, the decrease in viral replication observed when the functionality of L and 2A is impaired seems to depend on the host cell type. In this study, we have demonstrated, for the first time, the reproduction of EMCV with the impaired functionality of both the L and 2A proteins.

### 3.3. Functional Inactivation of L, but Not 2A, Leads to Reduced Viral Genome Replication during Infection

Previously, it was reported that the absence of a functional L protein reduced EMCV genome RNA synthesis upon infection in HeLa cells [11]. This effect was linked to the role of the L protein in suppressing antiviral host responses, such as the activation of NF-κB signaling or interferon expression [24]. Therefore, due to the potential interaction of 2A with L [29], we aimed to determine whether deletion of the 2A protein coding region would have a similar effect on the accumulation of genome RNA. 

The quantification of viral genome copies during infection in BHK-21 cells showed that there were no differences between the WT, Zfmut, Δ2A, and Zfmut&Δ2A viruses. The accumulation curve of genomic RNA did not show any significant changes due to the L- or 2A-mutations (Figure 3A). These results are consistent with the experiment aimed at determining the viral yield of the studied viruses in BHK-21 cells, where all the viruses demonstrated similar replication cycles (Figure 2A). The data obtained indicate that the detected small plaque mutants containing a partial deletion of 2A in a monolayer of BHK-21 cells (Figure 1B) are not associated with changes in the efficiency of their reproduction.

While a single-cycle experiment in HeLa cells revealed a ~10-fold reduction in viral RNA load during infection with Zfmut and Zfmut&Δ2A at 4–8 h.p.i., the Δ2A mutant demonstrated no difference in the number of genomic RNA copies per cell in comparison to the WT infection (Figure 3B). Thus, the decrease in viral RNA in HeLa cells is solely dependent on L-protein inactivation. The previously discovered decrease in Δ2A viral yield in HeLa cells (Figure 2B) is evidently not associated with a reduced ability to synthesize the viral genome.

These results indicate that functional inactivation of the 2A protein does not affect viral replication efficiency in cases of infection in HeLa and BHK cells. Mutations in the L protein resulting in its inactivation, however, can negatively affect the accumulation of viral genomic RNA in some cell lines.

### 3.4. Infection with Both L- and 2A-Deficient EMCV Mutants Induces Caspase-Dependent Cell Death in Infected HeLa Cells

The L protein of EMCV was already proposed to counteract apoptosis in infected HeLa cells [10,11]. Moreover, the ability of the 2A protein to suppress apoptosis in BHK-21 and PK15 cell lines was also previously demonstrated [30,31]. However, since these anti-apoptotic functions of the security proteins were demonstrated by using different cell lines, little can be concluded about the cooperative role of the security proteins in counteracting the apoptosis of infected cells. Therefore, we aimed to study the involvement and possible cross-talk between the L and 2A proteins in abrogating apoptotic cell death under the same conditions. For that purpose, we infected three different cell lines with WT, Zfmut, Δ2A, and Zfmut&Δ2A viruses and analyzed the apoptotic signatures in these cells using a TUNEL assay and Western blot analysis of caspase-3 activation. The TUNEL assay is based on fluorescent labeling of DNA termini by terminal nucleotide transferase thus allowing the microscopy detection of chromosomal DNA degradation. Additionally, the cells were stained with propidium iodide (PI), which is a common method to detect plasma membrane rupture as a result of necrotic cell death. We used light and fluorescence microscopy to examine the morphological changes In cells and their nuclei after infection by staining cellular DNA with the intercalating dye Hoechst 33342.

In agreement with our previous results, HeLa cells infected with WT EMCV at the end of the replication cycle demonstrated such morphological changes [10] as the dramatic shrinkage of the cell cytoplasm and nucleus (Figure 4) and the permeabilization of the cellular membrane detected by the PI staining of the nuclei (Figure 5A). Degradation of DNA was not detected during the WT infection (Figure 5A). Moreover, the described changes were independent of the presence of the pan-caspase inhibitor QVD-OPH (QVD). In sum, these results suggest that infection of HeLa with WT EMCV leads to necrotic cell death.

The functional inactivation of the L protein by introducing two point mutations to the zinc finger domain was already described as leading to the apoptotic death of HeLa cells infected with a mutant virus [10,11]. In agreement with this report, we observed a typical apoptotic cell death accompanied by chromatin condensation with nuclear fragmentation and apoptotic body formation upon HeLa infection with the Zfmut EMCV mutant (Figure 4). Zfmut-induced DNA fragmentation, as judged by a positive TUNEL signal, could be inhibited by QVD-OPH (Figure 5B). 

The infection of HeLa cells with the Δ2A mutant demonstrated different features than those observed in the WT and Zfmut infections. At the final stages of the Δ2A viral cycle (12 h.p.i.), the cells rounded up and the cytoplasm shrank; however, no apoptotic bodies were formed, and the cells did not separate into parts (Figure 4). Moreover, based on the PI staining, Δ2A infection also led to a plasma membrane rupture (Figure 5C). The cell shape of the HeLa cells infected with Δ2A also differed significantly from the infection with WT virus (Figure 4). Despite the signs described above, which are commonly associated with necrosis, infection with Δ2A also led to chromatin condensation and the fragmentation of DNA, which was inhibited by QVD-OPH like in the case of Zfmut infection (Figure 5C). After additional treatment with QVD-OPH, membrane rupture analyzed by PI staining was inhibited, suggesting that the permeabilization of the cell membrane upon Δ2A infection (Figure 5C) is likely associated with caspase activity. The addition of the caspase inhibitor also led to a modification of the morphological changes ongoing in infected cells, which were not as drastic as those observed in the Zfmut+QVD-OPH case (Figure 4) [7]. 

Similar to the Zfmut, infection with the Zfmut&Δ2A also resulted in HeLa cells forming apoptotic bodies, accompanied by nuclear fragmentation (Figure 4) with strong chromatin condensation and DNA degradation (Figure 5D), which were inhibited by treatment with QVD-OPH. Moreover, the plasma membrane of the infected HeLa cells remained intact, as judged by the absence of PI staining in the nuclei (Figure 5D). 

The time point for the positive control of apoptosis activation in HeLa cells by STS was chosen (6 h) based on when the action of this inducer led to the development of apoptotic features. Longer incubation resulted in cells detaching from the substrate. Uninfected cells (Mock) were incubated for a time equal to the longest virus incubation presented (12 h.p.i., Figure 5E).

Furthermore, we conducted Western blot analysis on the lysates of infected HeLa cells to detect the active forms of caspase-3, which is a marker for apoptosis. 

We detected a strong caspase-3 activation in the lysates of the Zfmut-infected cells, judged by the presence of an active 17 kDa cleaved form at 6 h.p.i. (Figure 6, Zfmut). Similar to the Zfmut, at the early stage of the HeLa infection (6 h.p.i.) with the Δ2A mutant, a strong caspase-3 activation was observed (Figure 6, Δ2A). Activation of caspase-3 during the infection with the Zfmut&Δ2A was also observed; however, the accumulation of the active form, comparable in signal to previous cases, occurred at a later stage (8 h.p.i.) (Figure 6, Zfmut&Δ2A). It could be explained by the lower efficiency of Zfmut&Δ2A replication in HeLa compared to other studied viruses (Figure 2B). A weak signal of the active caspase-3 form was detected in WT-infected lysates at both time points, suggesting that the activation cascade does not expand during the infection (Figure 6, WT).

The treatment of infected cells with QVD-OPH resulted in inhibition caspase-3 cleavage into active 17 kDa form (Figure 6). Moreover, the ~20 kDa form of caspase-3 was detected in the presence of QVD-OPH, which may correspond to a nonfunctional alternative cleavage form of caspase-3, as also observed previously [11].

To summarize, the functional inactivation of the L and 2A proteins of EMCV leads to the development of caspase-dependent cell death of infected HeLa cells. However, the typical morphological signs of apoptosis appear only in the cases of L-mutants: Zfmut and Zfmut&Δ2A. Partial deletion of the 2A results in the morphologically unique but caspase-3-assotiated death of HeLa cells.

### 3.5. Infection with Both L- and 2A-Deficient EMCV Mutants Induces Caspase-Dependent Cell Death in Infected RD Cells

RD cells infected with the WT EMCV underwent shrinkage with strong nuclear condensation, which did not respond to the pan-caspase inhibitor QVD-OPH (Figure 7). The PI-staining of the nuclei of the WT-infected RD cells suggests that the cellular membrane became permeabilized. We did not detect any TUNEL signal during the WT infection, indicating the absence of DNA degradation (Figure 8A).

The infection of RD cells with the Zfmut and Δ2A mutants resulted in nuclear fragmentation (Figure 7) accompanied by DNA degradation (Figure 8B,C), which were both inhibited by the addition of QVD-OPH. 

Infection with the Zfmut&Δ2A had the same effect on human RD cells as those with Zfmut or Δ2A. Cell death was accompanied by nuclear fragmentation (Figure 7) and DNA degradation (Figure 8D). All changes were sensitive to the presence of QVD-OPH. 

The obtained results suggest that WT EMCV induces the necrotic death of RD cells. All the mutants with L and 2A functional deficiencies seem to induce caspase-dependent apoptosis in infected RD cells.

The time point for the positive control of apoptosis activation in RD cells by STS was chosen (6 h) based on when the action of this inducer led to the development of apoptotic features. Longer incubation resulted in cells detaching from the substrate. Uninfected cells (Mock) were incubated for a time equal to the longest virus incubation presented (12 h.p.i., Figure 8E).

### 3.6. Functional Inactivation of the 2A Protein, but Not L, Leads to Apoptosis in the BHK-21 Cell Line upon Infection with EMCV

The hallmarks of necrotic cell death were also observed in BHK-21 cells infected with WT EMCV. The cells underwent shrinkage and strong nuclear condensation, which did not respond to the pan-caspase inhibitor QVD-OPH (Figure 9). Infection also induced plasma membrane permeabilization, which led to DNA staining with PI (Figure 10A). The weak TUNEL signal, which was hard to distinguish from the background, indicated that chromatin degradation was likely to be at a low level during the WT infection of the BHK-21 cell line (Figure 10A). There were no morphological signs of apoptosis (formation of apoptotic bodies, fragmentation of nuclei) detected in BHK-21 cells upon infection with WT EMCV (Figure 9). Surprisingly, unlike the other studied cell lines, BHK-21 cells infected with Zfmut showed no signs of apoptosis. Instead, Zfmut-induced morphological cell changes are identical to WT-induced changes. The cytoplasm of infected BHK-21 cells shrank, the nuclei were condensed without any dividing (Figure 10), and the plasma membrane became permeabilized (Figure 10B), all of which were not affected by QVD-OPH. 

BHK-21 cells infected with the Δ2A EMCV mutant demonstrated cell and nuclei fragmentation at the late stage of infection (16 h.p.i.). High TUNEL signals attest to the chromatin degradation during the infection with Δ2A of BHK-21 cells (Figure 10C). All described changes were sensitive to caspase inhibition by QVD-OPH (Figure 9 and Figure 10C). 

The infection of BHK-21 cells with the Zfmut&Δ2A resulted in the fragmentation of nuclei and entire cells (Figure 9). Additionally, DNA degradation was detected through a TUNEL assay (Figure 10D). As in the case of the Δ2A infection, the caspase inhibitor suppressed all the described cell changes (Figure 9 and Figure 10D). The infection of BHK cells with Zfmut&Δ2A demonstrated features similar to Δ2A infection.

Here we have revealed that only the absence of active 2A EMCV protein could induce apoptotic cell death in infected BHK-21 cells. The L protein evidently does not have an influence on BHK-21 cell death during the infection.

Taken together, an absence of the 2A security protein leads to caspase-dependent and likely apoptotic cell death upon EMCV of all studied cell lines.

The time point for the positive control of apoptosis activation in BHK cells by STS was chosen (6 h) based on when the action of this inducer led to the development of apoptotic features. Longer incubation resulted in cells detaching from the substrate. Uninfected cells (Mock) were incubated for a time equal to the longest virus incubation presented (16 h.p.i., Figure 10E).

### 3.7. Simultaneous Inactivation of the L and/or 2A Security Proteins Leads to the Amelioration of Cell Pathology upon Inhibition of Apoptosis

Previously, we have demonstrated that infection with an EMCV mutant lacking a functional L protein drastically delays the development of cell pathology if apoptosis is inhibited [10]. Inspired by such observations, we aimed to determine whether cell pathology can be equally delayed by caspase inhibition if 2A or both security proteins are non-functional.

Infection with WT EMCV caused the death 50% of infected HeLa cells at 8 h.p.i., as judged by membrane permeabilization, and that process did not respond to caspase inhibition with QVD-OPH, suggesting necrotic cell death (Figure 11A). Zfmut induces apoptotic cell death instead of necrotic cell death. Infection with Zfmut also resulted in 50% cell death at 8 h.p.i., as indicated by TUNEL-detected DNA degradation (Figure 11B). Therefore, the absence of the L security protein changes the type of cell death but not the rate at which cell pathology develops. However, in the case of L-deficient EMCV infection, suppressing apoptosis prolongs the survival of infected cells. When HeLa cells were infected with Zfmut and treated with the caspase inhibitor QVD-OPH, 50% cell death was observed at 16 h.p.i., which is 8 h later than observed with the WT infection (Figure 11A). Of note, QVD-OPH did not affect Zfmut reproduction kinetics; therefore, the increase in cell survival cannot be attributed to changes in infection efficiency per se (Figure 11C).

Deletion of the 2A security protein delayed cell death even without any additional treatment. Based on PI and TUNEL staining, infection with Δ2A led to 50% cell death only at 12 h.p.i.—4 h later than infection with the WT virus (Figure 11A,B), which was even further delayed to 16–24 h.p.i. by caspase inhibition (Figure 11A). Similar to Zfmut, such caspase inhibition by QVD-OPH did not reduce Δ2A reproduction (Figure 11C), and the cell death delay could not be associated with the lower viral yield. Thus, the removal of a single security protein simultaneously with apoptosis inhibition could prolong the survival of the infected cells during the EMCV infection. 

Surprisingly, the simultaneous inactivation of both security proteins did not further delay cell death. Similar to the Δ2A mutant, infection with Zfmut&Δ2A led to 50% cell death at 12 h.p.i., which was prolonged to 16–24 h.p.i. upon apoptosis inhibition in the presence of QVD-OPH (Figure 11A,B).

Our experiments with BHK cells also revealed the same phenomenon. When we partially deleted the 2A-coding region, which resulted in the apoptotic cell death of infected cells, and treated them with the caspase inhibitor QVD-OPH, we observed a delay in cell death (Figure 9 and Figure 10C,D).

### 3.8. Security Protein 2A Is Not Involved in Disruption of the Nucleo-Cytoplasmic Traffic

In addition to its role in counteracting apoptosis, security protein L has been shown to have another function in permeabilizing the nuclear envelope during EMCV infection [20,21]. However, it is not clear whether 2A is involved in this process as an independent factor or in cooperation with security protein L. To investigate the involvement of 2A in this process, we used a HeLa cell line with the NLS fused to three copies of enhanced green fluorescent protein (3xEGFP-NLS)—HeLa-3E [36]. As previously demonstrated, disruption of the nuclear pore complex by the cardiovirus L protein impairs nuclear-cytoplasmic transport, which can be detected by a loss of nuclear localization of 3xEGFP-NLS. Therefore, to assess the involvement of the 2A protein in disrupting nuclear-cytoplasmic transport, HeLa-3E cells were infected with WT, Zfmut, Δ2A, and Zfmut&Δ2A EMCV variants and analyzed using fluorescent microscopy at 4 and 6 h.p.i for 3xEGFP-NLS distribution (Figure 12).

As previously reported [20,21], WT EMCV infection caused 3xEGFP-NLS to relocate to the cytoplasm at 4 h.p.i., while infection with the L-deficient mutant did not result in cytosolic localization of 3xEGFP-NLS (Figure 12, Zfmut). However, deficiency in 2A did not prevent the relocation of 3xEGFP-NLS into the cytosol. Instead, as with the WT infection, 3xEGFP-NLS was detected in the cytoplasm at both 4 and 6 h.p.i. (Figure 12, Δ2A). Infection with Zfmut&Δ2A, lacking both security proteins, resulted in 3xEGFP-NLS retaining its nuclear localization at 4 h.p.i., similar to the Zfmut (Figure 12, Zfmut&Δ2A). While some cells infected with Zfmut and Zfmut&Δ2A showed the cytoplasmic localization of 3xEGFP-NLS at 6 h.p.i. (marked by arrows in Figure 12), this likely occurred as a result of the later stages of apoptosis, which have been shown to induce the degradation of the nuclear pore complex by apoptotic caspases [39]. Indeed, these cells demonstrated strong chromatin condensation and nuclear division, which are common signs of the later stages of apoptosis.

In summary, we have demonstrated that the 2A security protein is not involved in the disruption of nuclear-cytoplasmic traffic during EMCV infection and that this process is attributed to the L security protein.

## 4. Discussion

Picornaviral security proteins (L and 2A) are some of the most interesting virus-related objects for the exploration of the interaction between hosts and picornaviruses. Among the representatives of the *Picornaviridae* family, L and 2A are known to be proviral tools, the role of which is generally to act against multiple cellular reactions aimed at reducing the multiplication and spread of a virus [8]. The most common and adequate strategy for studying the properties of these proteins is the design of viruses that are defective in L and/or 2A functionality [11,23,30]. We used mutants of EMCV with nucleotide changes that destroy the well-known structural and functional domains of L and 2A proteins. We used substitutions (Cys19 → Ala and Cys22 → Ala) in the zinc finger domain responsible for known functions [11,21,24,40] in order to «switch off» L (Figure 1A). We used the L mutant (Zfmut) with amino acid substitutions instead of the mutant having a deletion in amino acid residues 12 through 52 of the L protein (ΔL) [23] due to the greater reproduction ability of the Zfmut [10]. We introduced a vast deletion in the 2A, containing from 11 to 124 residues (Figure 1A), aimed at removing the whole known structured part of the protein [33], named beta-shell, and saving the terminal parts, including the N-terminal 3C-mediated proteolytic cleavage site and NPG-P motif, that are responsible for the processing of the polyprotein [41]. It should additionally be noted that the YxxxxLΦ motif (129–135 aa), earlier predicted as an eIF4E-binding sequence mimicking the 4E-BP functional motif [38], remained intact in Δ2A and Zfmut&Δ2A mutants. However, current data on the structure of the 2A protein indicate that this site does not interact with eIF4E, due to structural differences between it and the eIF4E-binding domain of 4E-BP [42].

Several studies have defined the fact that L or 2A proteins are not directly involved in the replication process; mutants of the EMCV with dysfunctional L or 2A remain viable [11,24,30,43]. For the first time, we obtained a virulent EMCV mutant without both the L and 2A functional proteins. Lacking both known security proteins, the Zfmut&Δ2A mutant preserves its ability to reproduce in all studied cell cultures (Figure 1 and Figure 2). Moreover, it shows the unattenuating ability to produce infectious virus particles in BHK-21 cells. We can explain this surprising behavior by considering the peculiarities of the cell line. BHK-21 evidently has a defective signaling pathway started by RIG-1 [44], an ssRNA sensor that involves the antiviral innate immune system [45], which results in the inhibition of interferon production during a viral infection. Probably, thanks to the described features, BHK-21 cells become a «comfortable host» for mutants like Zfmut, Δ2A, and Zfmut&Δ2A, which have decreased reproduction ability in HeLa and RD (Figure 2B,C). There is no need for security proteins in order to reproduce effectively in a host with a defective antiviral response. To summarize, our experiments evidenced that both proteins of EMCV, L and 2A, could be inactivated while saving viral viability. The reproduction depression of all the mutants in HeLa and RD cells (Figure 2B,C) is evidence of the important role of both L and 2A in ensuring a high level of viral replication. Generally, it seems to be the main aim of all activities of the security proteins [8]. The reduction in viral replication with the dysfunction of 2A should be considered in more detail. When HeLa cells are infected with the Δ2A mutant, a decrease in virus yield is detected (Figure 2B), but it is not associated with a decrease in viral genome replication (Figure 3B). We have previously shown that in these cells the translation of viral proteins in the Δ2A mutant is also unchanged [46]. At the same time, we showed that in the case of this mutant, the final processing of capsid proteins is disrupted [46]. It should be noted that this phenomenon was noticed earlier [28]. It can be assumed that 2A somehow affects the processing of capsid proteins; when 2A is partially removed, this processing is disrupted, which can lead to a decrease in the yield of mutant viruses. However, this assumption does not explain the equal yield of all studied viruses in BHK-21 cells, so this aspect requires additional research. 

We detected an intriguing feature of Δ2A mutants—a small plaque phenotype in the BHK-21 cell monolayer (Figure 1B)—that has particular importance in terms of the significance of the type of cell death that finalizes the infection. Plaques formed in a monolayer of cells are areas of dead cells as a result of infection. During the experiment, the virus can only spread between neighboring cells since further transfer of the virus is blocked by the agarose coating. Therefore, the small size of plaques in comparison with other studied viruses, theoretically, may be associated with a reduced yield of the virus in cell culture or/and with a depressed ability to enter or/and exit cells. We have shown the independence of the size of plaques on BHK-21 cells from the characteristics of virus reproduction since all the studied mutants had the same yield. The ability of various viral mutants to enter most likely should not differ, since the virus capsid is the same in all the mutants. When infected with WT and Zfmut, BHK-21 cells undergo necrotic death, which is characterized by a disruption of the cell membrane (Figure 10A,B) and the release of cellular contents into the extracellular space. While infection with Δ2A and Zfmut&Δ2A leads to apoptosis, and the membrane did not lose integrity (Figure 10C,D). It is very likely that the type of cell death determines the size of plaques in the described case. Moreover, previously, a decrease in the viral release of the Δ2A mutant from BHK cells had been identified [30].

Apoptosis is one of the common cellular responses that activates during viral infection and is aimed at reducing viral-induced inflammation and viral spread by eliminating infected cells [47]. In recent decades, there have been indications that protein L of cardioviruses, specifically EMCV and TMEV, plays a significant role in antiapoptotic activity [11,48]. It is interesting to note that EMCV can inhibit chemically induced apoptosis in HeLa cells and evidently does so due to the activity of the L [11]. Moreover, published data showed that the deletion of 2A results in the apoptosis of EMCV-infected BHK-21 cells [30]. 2A may inhibit the chemical-induced apoptotic death of PK15 and BHK-21 cells via its ectopic expression [31]. All the described investigations included the study of one protein in a particular cell culture. We examined the properties of both proteins together in several cell lines at once. 

Table 1 contains the signs of cell death after the infection of HeLa, RD, and BHK-21 cells with WT, Zfmut, Δ2A, and Zfmut&Δ2A viruses. In the case of RD cells, the removal of any of the proteins apparently led to the development of apoptosis. We did not see any differences in the death of RD cells during infections with mutants. Thus, both L and 2A exhibit antiapoptotic properties in RD.

The L protein has previously been shown to exhibit antiapoptotic properties in HeLa cells [11]. L dysfunction leads to the development of apoptosis in infected cells, accompanied by typical morphological and molecular signs: cell division into bodies, nuclear fragmentation, DNA degradation, and caspase-3 activation (Figure 4, Figure 5B and Figure 6). Moreover, the described changes occur regardless of the presence of functional 2A. In turn, the partial deletion of protein 2A leads to the special death of HeLa cells, the signs of which are described in Table 1. The nuclei of infected cells condense without fragmentation, which is accompanied by DNA degradation, and the integrity of the cell membrane is disrupted (Figure 4 and Figure 5C). Despite the presence of apoptotic program development signs, such as the activation of caspase 3 (Figure 6), other signs indicate the possible development of pyroptosis [16]. The activation of caspase-3 we observed during Δ2A infection probably indicates a non-canonical pathway for the development of signs of pyroptosis [49,50]. Caspase-3 can trigger the process of disrupting the integrity of the cell membrane by activating gasdermins, in particular, DFNA5 [51]. We probably detected the result of a similar process by staining cell nuclei with PI during infection with the HeLa mutant Δ2A, which was sensitive to the addition of QVD-OPH (Figure 5C). It is noteworthy than an additional mutation of the L protein introduced in the Zfmut&Δ2A leads to the development of classical apoptosis (Figure 5D). This fact suggests that pyroptosis-like cell death during infection with Δ2A of HeLa cells could develop due to the activity of L, which might inhibit the development of the blebbing of membranes and the formation of apoptotic bodies. To summarize, both proteins L and 2A appear to act as inhibitors or modulators of the caspase-dependent death of infected HeLa cells through apoptosis or, potentially, pyroptosis. However, the detailed mechanism of death and the proper role of L and 2A in it will be studied by us in the future. In this work, we showed that depending on the combination of EMCV L and 2A proteins, several outcomes of cell death are possible. Morphologically, the death of the same cell type may differ from apoptosis and necrotic CPE.

As we noted, apoptotic death in HeLa cells during infection with the L-deficient mutant develops in the presence or absence of full-sized 2A. This fact suggests that these two proteins may influence different cell death program activation pathways. A strong argument in favor of this assumption is the results of experiments on the BHK-21 cell line. We have shown, for the first time, that in some cell lines, like BHK-21, the presence of the well-known antiapoptotic L protein does not influence a cell death pathway (Figure 9 and Figure 10). Based on the obtained results, we may conclude that only protein 2A could influence the apoptotic pathway in BHK-21 during the infection. We can interpret that phenomenon based on the peculiarities of the cell line. BHK-21 is a cell line defective in the RIG-1 pathway of interferon activation [44]. This pathway is closely related to apoptosis during RNA-viral infection due to double-RNA sensing by RIG-1 and interferon-mediated activation of the mitochondrion route of apoptosis [52]. The L protein of cardioviruses acts as an inhibitor of interferon expression and activation of interferon-related genes like IRF-3 and NF-κB [24,26], which also take part in a RIG-1-driven interferon activation. So, all the facts indicate that L protein does not play any role in an EMCV-infected BHK cell fate due to the possible absence of one of the apoptotic signaling pathways, that L can probably modify. 

The mechanism of 2A interference in cell death is still unclear. There is only one experimentally based assumption about mechanism: EMCV 2A plays an antiapoptotic role by interaction with Annexin A2 [31]. Nevertheless, this proposition is based on a 2A-expressing model, not a viral-infecting model. Furthermore, Annexin A2 has a large number of cellular functions [53], and it is too far to fully understand how the interaction can change the death fate. The discovery of a striking feature of cardiovirus 2A in recent years has likely shed light on a possible mechanism by which 2A influences cell death. 2A acts as a trans-activator for −1 frameshifting inside the 2B-coding region, which makes it a regulator of downstream protein synthesis [32]. Structure resolution of the 2A discovered part of the protein that is responsible for that process [42], the arginine loop (95–100 a.a.), which was deleted in our mutants Δ2A and Zfmut&Δ2A. During the -1 frameshifting that occurs in the middle of the viral cycle (6–8 h.p.i., L929 cells), the 2B* protein translates on the alternative frame [2], and non-structural proteins 2B, 2C, 3AB, 3C, and 3D synthesize with less efficiency. Contrariwise, during non-frameshifting conditions, such as Δ2A infection, 2B* does not form and proteins from 2B to 3D express with greater efficiency compared to WT infection [32,46]. Therefore, 2A may alter the cell death pathway indirectly by changing the ratio of viral proteins. Apoptosis or another programmed cell death could potentially be activated during the Δ2A infection via different reasons connected with some viral proteins. The increase in the synthesis of 3C/3CD protease can overdegrade cellular proteins [54]. The increased accumulation of the 2B protein, which seems to be a viroporin, might induce Ca^2+^ flux from the Golgi apparatus and stimulate NLRP3 inflammation [55]. Moreover, the functional features of the frameshifting-born 2B* protein are still unknown [2]. It cannot be ruled out that 2B* may play an antiapoptotic role, and its synthesis shut-off during the Δ2A infection might be the cause of apoptotic development. These assumptions require confirmation or refutation in the future. Nevertheless, 2A seems to be a cell death regulator during the infection, direct or indirect.

The results of experiments in HeLa, RD, and BHK-21 cells suggest that L and 2A obviously both take part in the regulation of infected cell death and might do so through different routes. On the basis of the presently reported and previous data, we modify the model [10], implying that infection with EMCV may induce suicidal processes possessing several branches: necrotic, apoptotic, and presumably pyroptotic. We assume that there may be more possible branches. The choice between the development of one or another pathway depends not only on the presence of functional proteins L and 2A, but also on the characteristics of the host cells. 

Previously, we have demonstrated the possibility of delay in the development of pathology in infected cells by the simultaneous inactivation of the EMCV L protein and inhibition of apoptosis development [10]. Here we presented that EMCV disarmament via partial deletion of the second security protein 2A together with cellular disarmament via inhibition of apoptosis by QVD-OPH also results in the amelioration of pathology during the infection of HeLa (Figure 11), BHK-21 (Figure 9 and Figure 10), and RD (Figure 7 and Figure 8) cells. The appearance of the morphological signs of death of these cells (in the form of DNA degradation or membrane permeability) is clearly delayed. At the same time, in the presence of QVD-OPH, the mutant’s reproduction cycle was not depressed (Figure 11C). Thus, the appearance of a delay in cell death is not associated with changes in the viral cycle, but directly depends on changes in the development of death processes due to the action of inhibitors (QVD-OPH) and changes in the functionality of security proteins. In addition to our previous investigation [10], here, we obtained data that points out the universality of the possibility of suppressing cellular pathology via the functional violation of both virus security proteins and the inhibition of cell death. At least that phenomenon does not depend on the type of cell or certain viral proteins. Moreover, we have once again shown that the process of the development of cellular pathology can be uncoupled from the propagation of a lytic virus; the accumulation of infectious viral particles may not always cause rapid cell death. To delay cell death or, potentially, abolishment, it is necessary to influence the security proteins of the virus and the cell death program.

Here we showed the seemingly invisible properties of EMCV security proteins which call into question one of the generally accepted theories. It is believed that L and 2A likely interact with each other [29]. It has even been suggested that protein 2A, having a potential signal for nuclear localization [38], transports L into the nucleus or near the nucleus of the cell to carry out its functions. Thus, disruption of 2A function, for example, by deleting a potential L-binding site [29] or a nuclear localization signal (both deleted in the Δ2A mutant), would affect L functionality. The L protein of cardioviruses is a well-known viral «tool» for disturbing nuclear cytoplasmic traffic [19,20,21], that appears to play a critical role in suppressing the cell’s antiviral immune response. Our experiments with L- and 2A-mutants of EMCV have shown that 2A does not play any detectable role in this process. Removal of the 2A functional site did not affect the “work” of L, which performed its function during the Δ2A infection (Figure 12). Another known function of L is to counteract the interferon system. Violation of the functionality of L leads to well-known consequences: a decrease in the efficiency of the accumulation of genomic RNA [24,56] (Figure 3B). Presumably, the suppression of 2A functionality should have suppressed the effect of L, reducing the efficiency of viral RNA accumulation during Δ2A infection; however, this did not happen. The accumulation of Δ2A viral RNA turned out to be the same as in the WT (Figure 3B). These two facts cast doubt on the validity of the model of interaction between L and 2A.

Since the L and 2A proteins are involved in cell death modification, they play an incredibly important physiological role during viral infection. The data obtained by us and other researchers show that the functioning of these proteins as part of the WT virus in infected cells leads, as a rule, to their necrotic death. This promotes effective viral spread through the destroyed cell membrane. Apparently, the rate of the development of death processes also depends on the functioning of L and 2A. Probably, by influencing the functional centers of these proteins with the help of chemicals, it will be possible to control the pathological processes that the infection causes. For instance, by inducing apoptosis instead of necrosis by the chemical inhibition of the security protein, one will reduce viral spread in the body and activate specific immunity. That is why the study of proteins like L and 2A is, in our opinion, one of the most important tasks of modern virology.

## Figures and Tables

**Figure 1 viruses-16-00280-f001:**
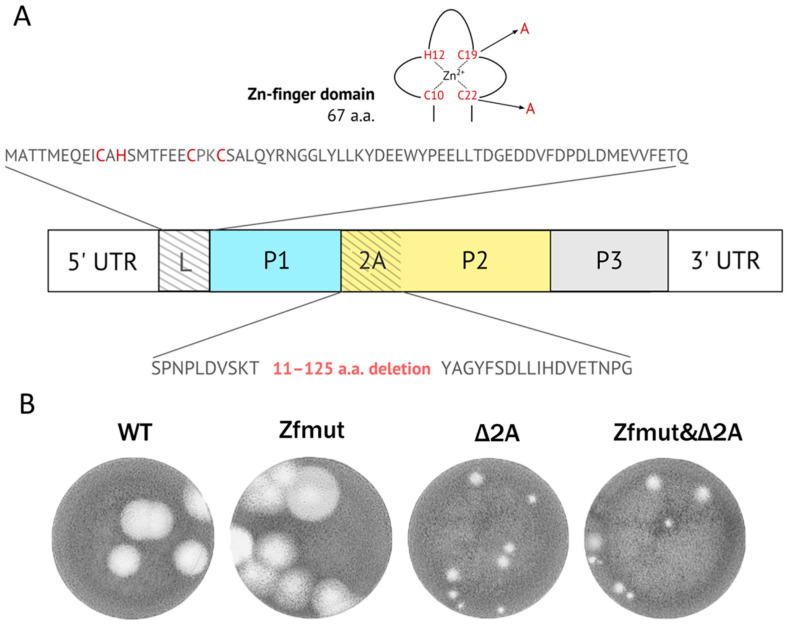
(**A**) Genome diagram of EMCV and modifications introduced in Zfmut, Δ2A, and Zfmut&Δ2A mutants. (**B**) Plaque phenotype of obtained pools of WT, Zfmut, Δ2A, and Zfmut&Δ2A viruses in a BHK-21 monolayer after 4-day incubation.

**Figure 2 viruses-16-00280-f002:**
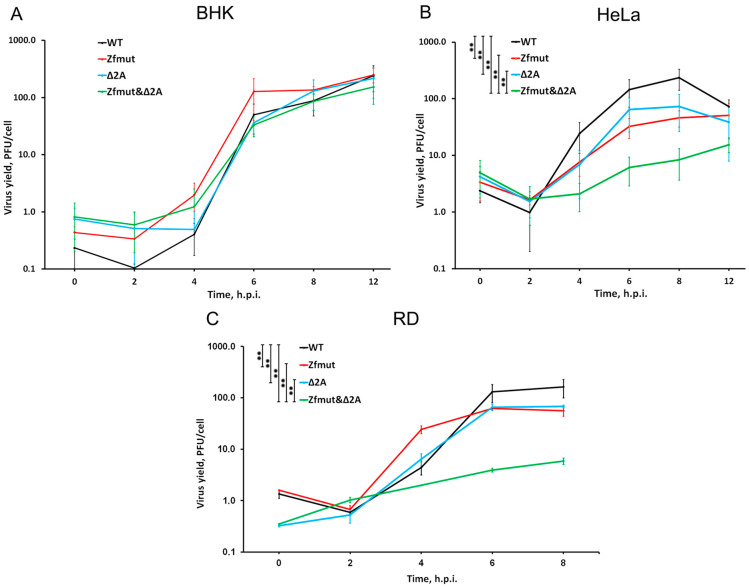
Accumulation of infectious viral units during single-cycle reproduction of WT, Zfmut, Δ2A, and Zfmut&Δ2A EMCVs in (**A**) BHK-21, (**B**) HeLa, and (**C**) RD cells. ** *p* < 0.01.

**Figure 3 viruses-16-00280-f003:**
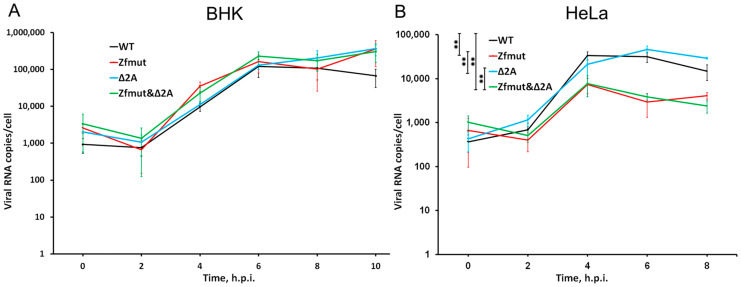
Accumulation of viral genomic RNA during single-cycle reproduction of WT, Zfmut, Δ2A and Zfmut&Δ2A EMCVs in (**A**) BHK-21 and (**B**) HeLa cells. ** *p* < 0.01.

**Figure 4 viruses-16-00280-f004:**
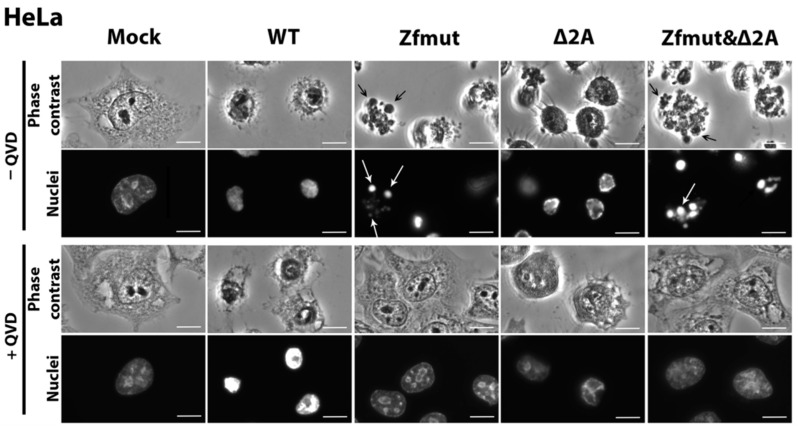
Morphological and nucleolar changes during the infection of HeLa cells with WT, Zfmut, Δ2A, and Zfmut&Δ2A EMCVs at 12 h.p.i. Hoechst 33342 staining of nuclei. Mock—uninfected cells. QVD—20 µM. Bars correspond to 10 µm. Black arrows indicate apoptotic bodies, white arrows indicate nuclear fragments.

**Figure 5 viruses-16-00280-f005:**
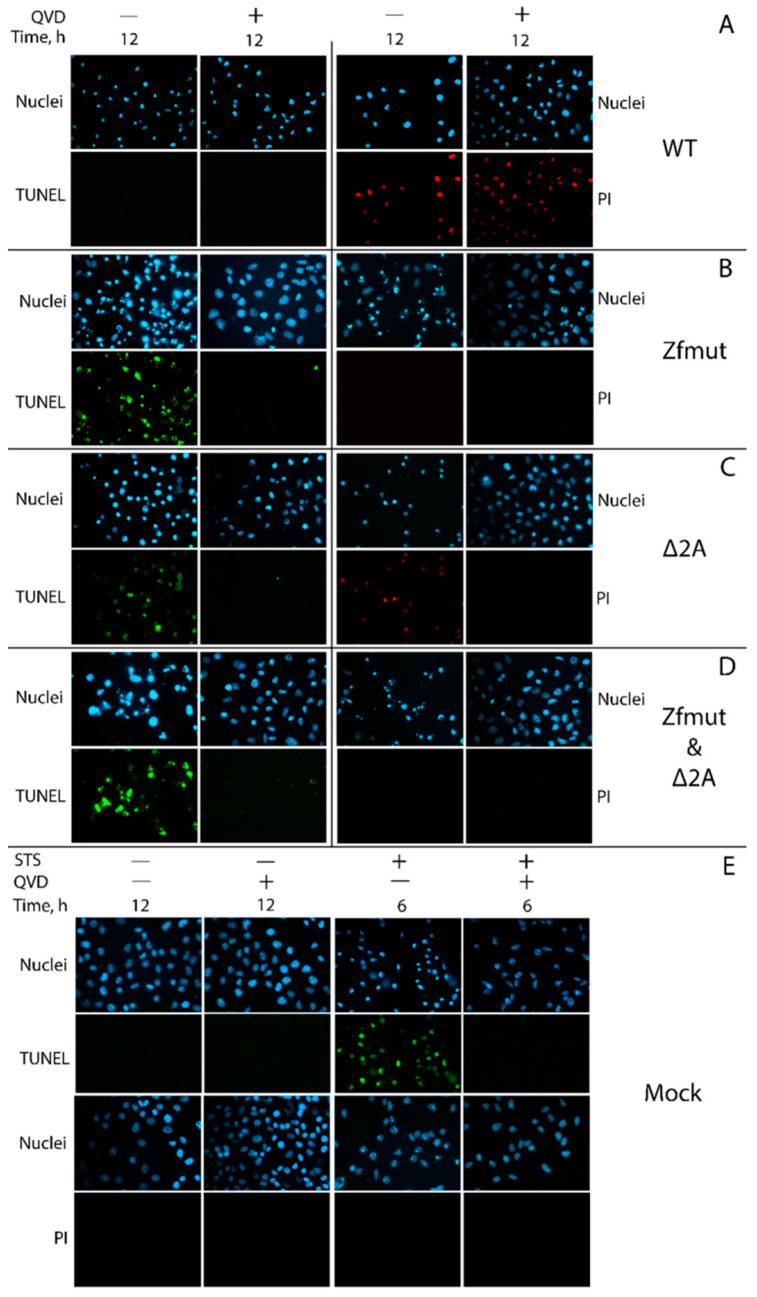
DNA fragmentation detected by TUNEL staining and cellular membrane permeabilization detected by propidium iodide staining in HeLa cells during the infection with WT (**A**), Zfmut (**B**), Δ2A (**C**), and Zfmut&Δ2A (**D**) EMCVs. MOI: 40 PFU/cell. QVD, 20 µM. STS, 500 nM. Nuclei stained with Hoechst 33342. Mock (**E**)—uninfected cells. PI—propidium iodide. TUNEL-positive signal was detected during Zfmut, Δ2A, and Zfmut&Δ2A infection and STS treatment in the absence of QVD-OPH. PI-staining of nuclei was detected during the WT-infection regardless of QVD and the Δ2A-infection in the absence of QVD-OPH.

**Figure 6 viruses-16-00280-f006:**
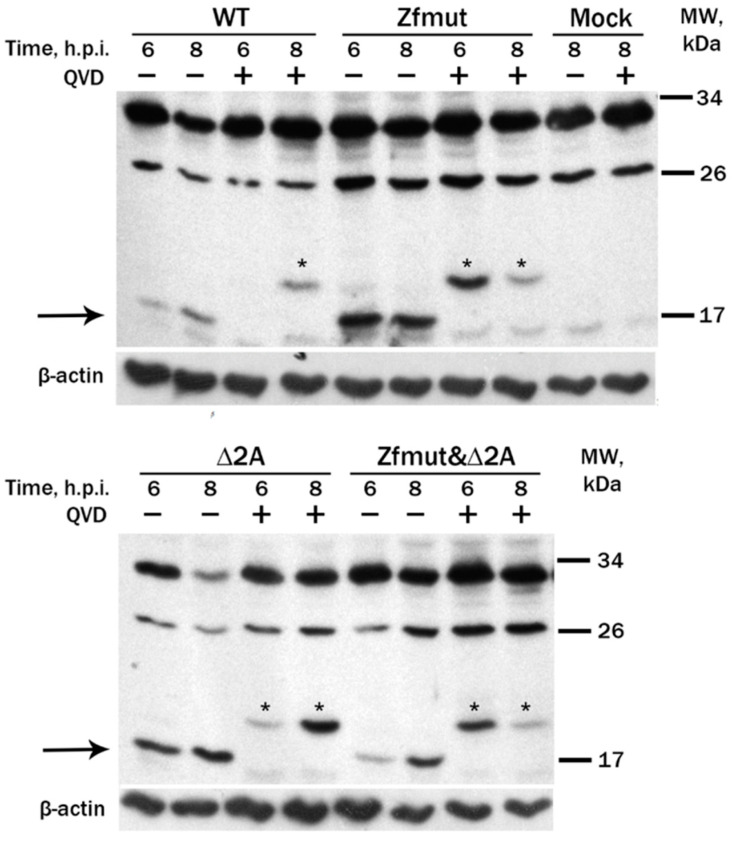
Western-blot analysis of partial proteolysis of caspase-3 in HeLa cells during the infection with WT and Zfmut, Δ2A, and Zfmut&Δ2A EMCVs. MOI—40 PFU/cell. QVD—20 µM. Mock—uninfected cells. Processed caspase-3 bands are marked with arrows. Slow-migrating bands of abnormally processed caspase-3 bands in the presence of QVD are marked with asterisks (*).

**Figure 7 viruses-16-00280-f007:**
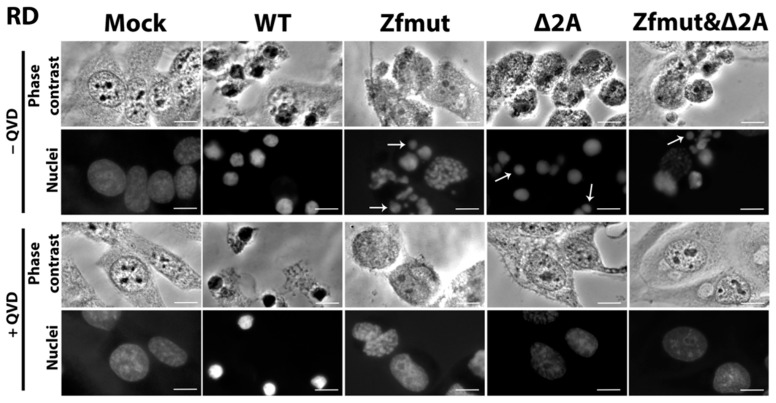
Morphological and nucleolar changes during the infection of RD cells with WT, Zfmut, Δ2A and Zfmut&Δ2A EMCVs at 12 h.p.i. Hoechst 33342 staining of nuclei. Mock—uninfected cells. QVD—20 µM. Bars correspond to 10 µm. White arrows indicate nuclear fragments.

**Figure 8 viruses-16-00280-f008:**
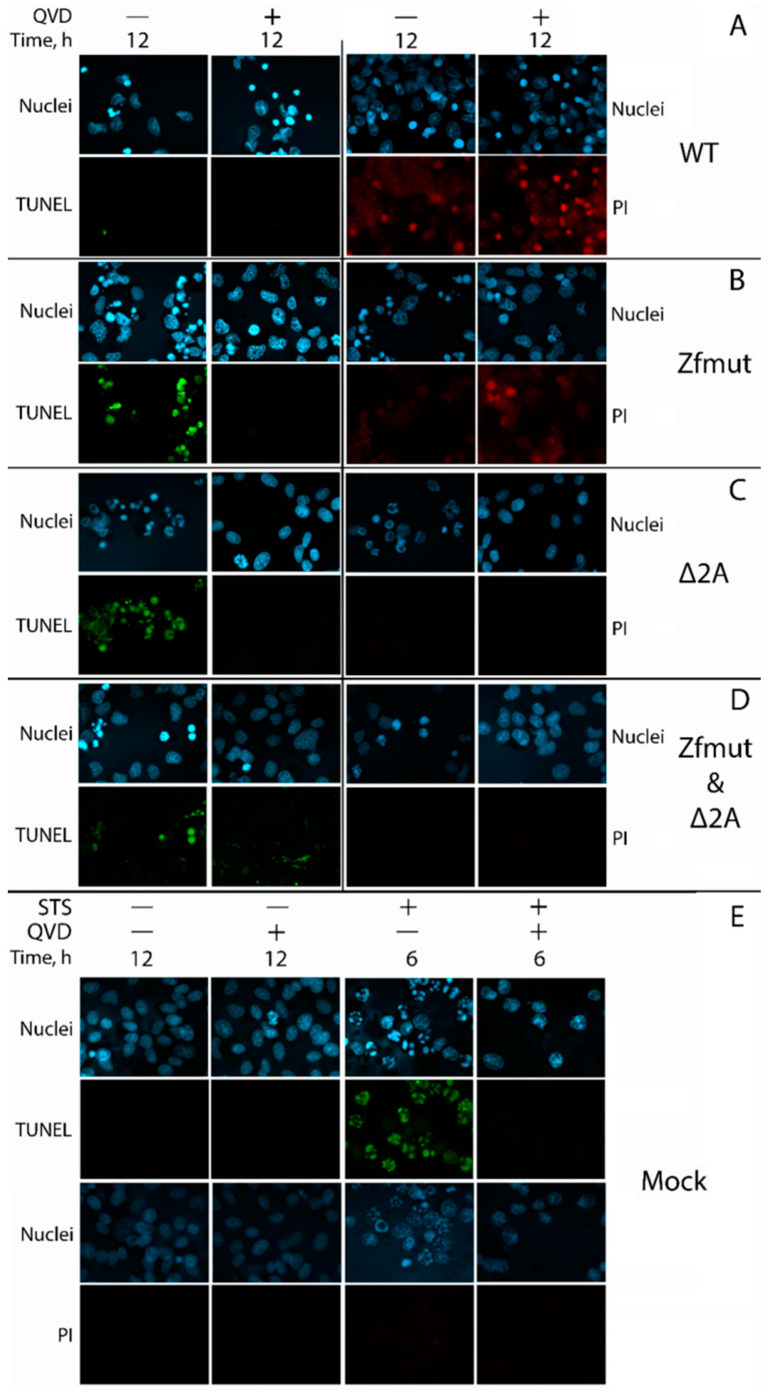
DNA fragmentation detected by TUNEL staining and cellular membrane permeabilization detected by propidium iodide staining in RD cells during the infection with WT (**A**), Zfmut (**B**), Δ2A (**C**), and Zfmut&Δ2A (**D**) EMCVs. MOI: 40 PFU/cell. QVD—20 µM. STS—500 nM. Nuclei stained with Hoechst 33342. Mock (**E**)—uninfected cells. PI—propidium iodide. TUNEL-positive signal was detected during Zfmut, Δ2A, and Zfmut&Δ2A infection and STS treatment in the absence of QVD, PI-staining of nuclei was detected during the WT-infection regardless of QVD and the Zfmut-infection in the presence of QVD.

**Figure 9 viruses-16-00280-f009:**
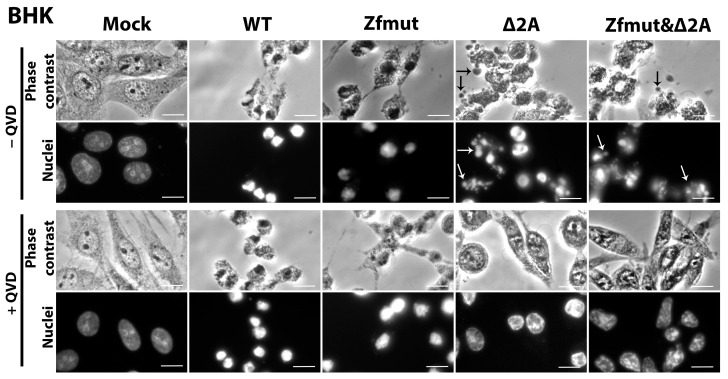
Morphological and nucleolar changes during the infection of BHK cells with WT, Zfmut, Δ2A, and Zfmut&Δ2A EMCVs at 16 h.p.i. Hoechst 33342 staining of nuclei. Mock—uninfected cells. QVD—20 µM. Bars correspond to 10 µm. Black arrows indicate apoptotic bodies, white arrows indicate nuclear fragments.

**Figure 10 viruses-16-00280-f010:**
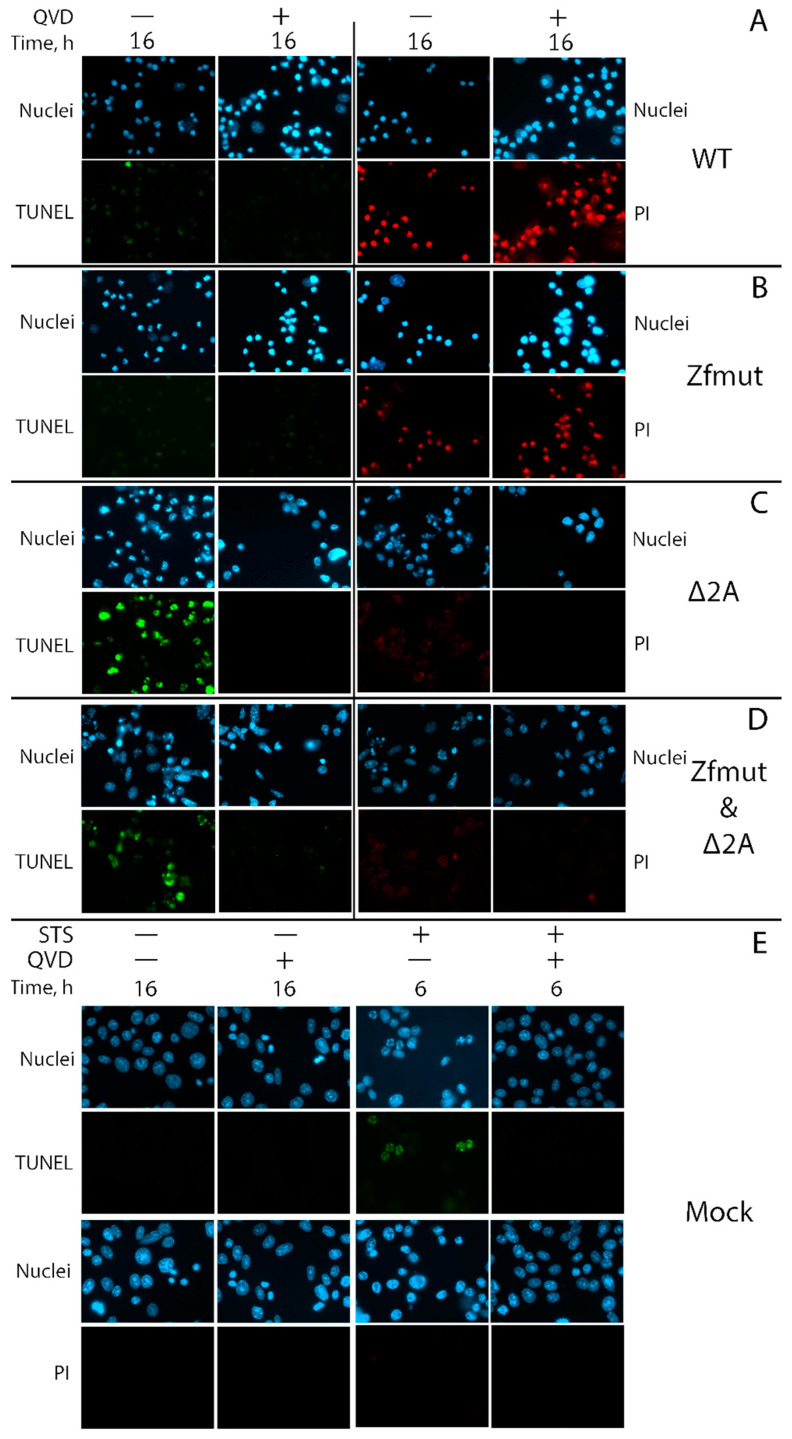
DNA fragmentation detected by TUNEL staining and cellular membrane permeabilization detected by propidium iodide staining in BHK cells during the infection with WT (**A**), Zfmut (**B**), Δ2A (**C**), and Zfmut&Δ2A (**D**) EMCVs. MOI—40 PFU/cell. QVD—20 µM. STS—500 nM. Nuclei stained with Hoechst 33342. Mock (**E**)—uninfected cells. PI—propidium iodide. TUNEL-positive signal was detected during Δ2A and Zfmut&Δ2A infection in the absence of QVD, PI-staining of nuclei was detected during WT and Zfmut infection regardless of QVD.

**Figure 11 viruses-16-00280-f011:**
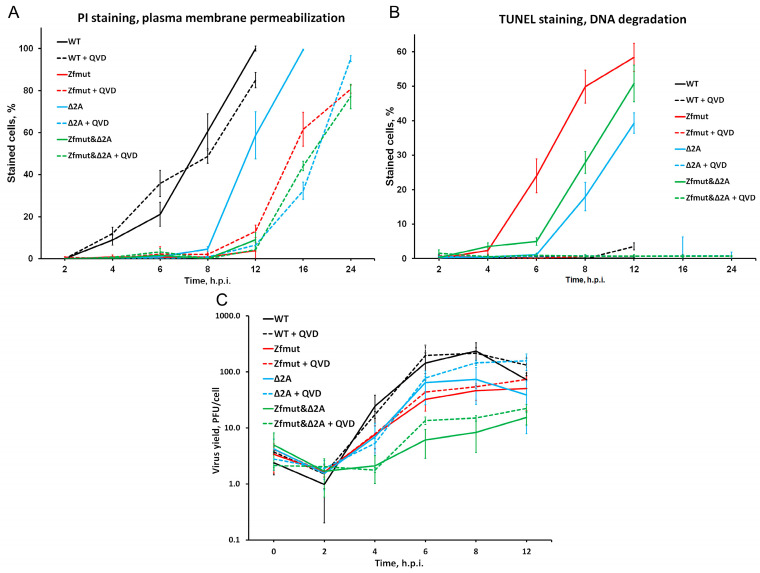
Kinetics of propidium iodide (PI) (**A**) and TUNEL staining (**B**) of HeLa cells during the infection with WT and Zfmut, Δ2A, and Zfmut&Δ2A EMCVs. (**C**) Accumulation of infectious viral units during single-cycle reproduction in HeLa cells of mutants in the presence of QVD (20 µM).

**Figure 12 viruses-16-00280-f012:**
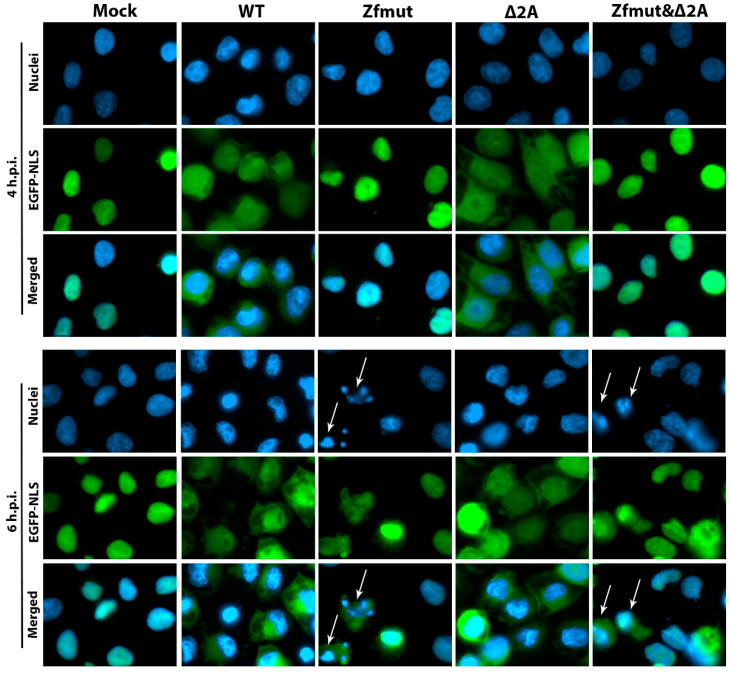
Distribution of 3xEFGP-NLS inside HeLa-3E cells during the infection with WT and Zfmut, Δ2A and Zfmut&Δ2A EMCVs. Nuclei stained with Hoechst 33342, mock—uninfected cells. Cells with the redistribution of 3xEGFP-NLS due to finalized apoptosis are marked with arrows.

**Table 1 viruses-16-00280-t001:** Synoptic cell death description *.

Cell Line	WT	L-mutant (Zfmut)	2A-mutant (Δ2A)	L&2A-mutant (Zfmut&Δ2A)
	Cell Death Description			
HeLa	Membrane permeabilization, shrinking of nuclei, caspase independent	DNA degradation, fragmentation of nuclei, formation of apoptotic bodies, caspase dependent	DNA degradation, membrane permeabilization, shrinking of nuclei, caspase dependent	DNA degradation, fragmentation of nuclei, formation of apoptotic bodies, caspase dependent
RD		DNA degradation, fragmentation of nuclei, caspase dependent		
BHK-21	Membrane permeabilization, shrinking of nuclei, caspase independent		DNA degradation, fragmentation of nuclei, formation of apoptotic bodies, caspase dependent	

* Features of necrotic cell death are marked in black, apoptotic—in red, and pyroptotic—in blue.

## Data Availability

Data are contained within the article.

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
