# Peer review of "Comprehensive Elucidation of the Role of L and 2A Security Proteins on Cell Death during EMCV Infection"

_viruses, 2024, doi:10.3390/v16020280_

Round 1
Reviewer 1 Report (Previous Reviewer 4)
Comments and Suggestions for Authors
The authors present a thoroughly revised manuscript.
1. According to my understanding of the picornavirus plaque assay, plaque sizes reflect replication efficiency. All plaques start with a single infected cell, and virus spreads after release by infection of neighboring cells; a large plaque indicates more lysed cells at day 4 p.i. than a small plaque. If this concept of plaque assay is true, why do mutants D2A and Zfmut&D2A yield small plaques (Fig. 1B) but high virus titres (Fig. 2A) and high values of RNA copies (Fig. 3A) in BHK cells? From the plaque assay, I would have expected a reduced virus titre and reduced values of RNA copies. The authors offer two explanations and I disagree with both. First, the authors' hypothesis that just "accumulation of viral genomic RNA" is affected. What is meant with accumulation of RNA? Something like degradation of newly synthesized RNA? Second, the hypothesis of impaired capsid protein processing is also unsatisfying.
2. Discussion is rather lengthy. Shorten it!
3. line 34: "it's" should read "its"
4. lines 34-35: The first EMCV strains were isolated from mice: Columbia SK virus (Jungeblut et al. 1942) and MM virus (Jungeblut and Dalldorf 1943), the first EMCV strain from a primate was isolated in 1945 from a chimpanzee in Florida (Helwig and Schmidt 1945), followed by Mengo virus from a rhesus monkey (Dick 1948). Another early EMCV strain is mouse Elberfeld virus of 1949 (see Franklin et al. 1959).
5. line 90: It should read "studies".
6. lines 119-120: "Due to its probable interaction..." This is a hypothesis and in the discussion section, the authors refute this hypothesis (compare lines 765-772). Rephrase!
7. line 128: What is DMEM, Russia?
8. lines 183-185: It should read "...for the indicated intervals, cells underwent necessary procedures..."
9. line 189-190: It should read "Viral growth experiments were replicated at least 3 times for all cell lines."
10. line 190: Explain abbreviation.
11. line 215: It should read "western blot analysis".
12. line 406, Fig. 6: The band at 34 kDa is likely unprocessed caspase-3. What protein is represented by the 26-kDa band?
13. line 409: It should read "abnormally processed".
14. line 601: "unattenuating viability" -> Is this true? The double mutant has a small plaque phenotype which indicates that one of several interstages in virus multiplication may be affected. Delayed lysis or another virus-releasing mechanism may explain small plaques. Infected cells, however, may harbour unreleased progeny virions and virus RNA.
15. line 634: The term "envelope" is used in virology to denote a lipid membrane surrounding a nucleocapsid. Authors may use a term like capsid or shell instead.
16. lines 637-642: Redundancy. Delete one sentence.
17. lines 679-681: It should read "...than an additional mutation of the L protein...".
18. lines 681-683: It should read "... could develop due to ...".
19. lines 687-689: "In this work, we showed..." Weaken this statement. Pyroptosis was not shown here. Pyroptosis is caspase-1 dependent. Authors showed activation of caspase-3 only and assumed non-canonical pyroptosis pathway.
20. line 726: Translate the Cyrillic phrase into English (reference 63 of some previous paper).
Comments on the Quality of English LanguageEnglish language has been improved. However, there are still few mistakes, mostly typos.
Author Response
Dear Reviewer, thank you for all comments.
Please see the attachment with answers.

Reviewer 2 Report (Previous Reviewer 2)
Comments and Suggestions for Authors
Thank you very much for your efforts to kindly answer my concerns. The authors answered properly to my concerns in the revised manuscript.
Author Response
Dear Reviewer!
Thank you for carefully reading our manuscript.
This manuscript is a resubmission of an earlier submission. The following is a list of the peer review reports and author responses from that submission.
Round 1
Reviewer 1 Report
Comments and Suggestions for Authors
Encephalomyocarditis virus (EMCV) is an important zoonotic pathogen that causes an acute infectious disease characterized by encephalitis, myocarditis, or perimyocardial inflammation in pigs and certain mammals, rodents, and even primates. The manuscript entitled " Comprehensive elucidation of the role of L and 2A security proteins on cell death during EMCV infection" described an interesting finding on function of L and 2A protein of EMCV on cell death of in three cell lines. It was found that both L and 2A are non-essential for viral reproduction in HeLa, BHK and RD cell lines. However, deletion of either protein leads to apoptosis of the infected cells. It was also demonstrated that functional inactivation of both L and 2A security proteins together with apoptosis inhibition leads to a delay in cell death progress. And L protein is involved in abrogation of nuclear cytoplasmic transport independently from the presence of full-length 2A protein. The study is interesting and the data are well presented with in-depth discussion. However, the manuscript needs English language revision.
Some comments are list below:
1. The manuscript requires major proof reading by a native speaker.
2. The introduction part is too lengthy and should more concise, only concise background closely related to the study is need. In addition, a brief introduction of EMCV, such as prevalence and its impact on humans or animals, should be given to make it easier for readers to understand the disease and the importance of this study.
3. Zinc-finger, instead of Zn-finger, is more commonly used. Please revised them in the manuscript.
4. The materials and methods section lacks an introduction to statistical methods, including the number of experimental replicates etc.
5. Line 223-233, “Funcionality of the Zn-finger domain was abolished……same protocol followed by plaque cloning procedure.” describes the mutants construction method and should be included in the “Materials and Methods” part (section 2).
6. Figure 1 should include color images indicating the growth of the recombinant virus on different cell lines (HeLa, BHK-21 and RD) used in this study.
7. All the Figures in supplemental part should be included in the main text. Such as S1 should be figure 2C, Figure S2 should be figure 4C, etc.
8. It is preferred to have a conclusion describing the findings and significances of the study at the end of the article.
Reviewer 2 Report
Comments and Suggestions for Authors
In this study, Ivin et al. analyzed the role of the leader protein and the 2A protein of EMCV in cell death after the infection, especially focusing on the simultaneous functional KO of these proteins. They observed the cell-type-specific role of these proteins in terms of cell death.
The results are generally straightforward and interesting; however, the relationship between observed phenomena seemed unclear and some integrated discussion to explain the relationship might be required. The observed difference between the L protein and the 2A might be clearly discussed in terms of the below points (e.g., apoptosis of BHK-21 cells infected with the Δ2A mutant). The addition of a new Table that summarizes the observations might be helpful for the readers.
Viral replication level (RNA, protein)
Membrane permeabilization
Suppression of interferon activation (RIG-I induced pathway of apoptosis)
Specific points:
- L112: A reference that showed 2Apro of PV is not essential for viral propagation (Igarashi et al., 2010, J Virol), despite various biological effects reported for the 2Apro, might be cited.
- L150: The exact condition of the plaque assay might be provided. It might not be clear which cell was used for virus propagation and for virus titration.
- L221, 525: Deletion of the L protein-coding region seemed more succinct than the destruction of the Zn-finger domain, such as the 2A-coding region. An explanation of this point might be helpful for the readers.
- Fig.2 and other figs.: Used cell lines might be described in the graphs as well as in the legends.
- L275: The potential role of the 2A in virion production might be discussed as the deletion did not affect the genomic RNA copies in HeLa cells.
- L326 and 371: Please clarify or show “blebbing” and apoptotic bodies in Figure 4A. The difference between HeLa cells and BHK-21 cells seemed not clear in terms of blebbing in this figure.
- L376: Chromatin condensation and fragmentation by the delta 2A in Fig.6 was not clear. Please clarify in the figure.
- Figs.5, 6, 8: The time for STS or mock treatment was different (8, 6, or 16 h). An explanation of this difference should be clarified in the text.
- Figures: The conclusion of the observation might be included in the figures (e.g., membrane rupture + or -, TUNEL + or -, etc.). It might be difficult for the readers to clearly see the authors’ conclusions in the data. For the interpretation, quantification of the data is desirable.
- L399-400, 413-415: These sentences seemed inconsistent. Please clarify.
- Fig.11: Merged images might be useful to clarify the transition of the localization.
- L535: replication?
- Some discussion on the mode of cell death and its physiological importance might be included.
Some typos were observed.
Reviewer 3 Report
Comments and Suggestions for Authors
In the manuscript entitled "Comprehensive elucidation of the role of L and 2A security proteins on cell death during EMCV infection", the authors analyzed the effect of the EMCV leader (L) and 2A proteins on cell survival. For this purpose, they generated functional defected mutants (Zfmut, Δ2A and Zfmut&Δ2A mutants). The authors could confirm results obtained by others that both proteins are involved in the regulation of cell survival. The main aspect of the manuscript has been already presented somewhere else (e.g. PMID: 21849462) and the novelty needs definitely to be increased for example by dissecting the precise molecular mechanism how both proteins contribute to the regulation of cell survival. Thus, the overall novelty of the here presented data is low and the manuscript should not be considered for publication in the widely accepted journal viruses.
Reviewer 4 Report
Comments and Suggestions for Authors
The authors describe their experiments conducted to compare the effects of wildtype and inactivated EMCV L and 2A proteins on virus replication efficiency and the cytopathogenic effect on the host cells. The effect of single inactivated proteins or a combination of both inactivated proteins was investigated in three cell lines, HeLa, GMK and RD cells. The authors aimed to study – for the first time – "the properties of L and 2A proteins in the same context".
General comments:
Each of the two proteins, L and 2A, has a multifunctional role in the replication cycle of EMCV. One of these functions is the inhibition of apoptosis. Hence, mutation of L or deletion of large parts of the 2A gene destroys this inhibitory function although slightly different effects may be seen in the various assays applied here. Abrogation of both proteins seems to be an unsuited tool to detect significant differences: inhibition of one anti-apoptotic mechanism leads to apoptosis; inhibition of two anti-apoptotic mechanisms leads also to apoptosis. What did I learn? To my opinion, the authors failed to present convincing data which demonstrate a possible interaction or cooperation of L and 2A proteins. May be that L and 2A affect different apoptotic pathways – the experiments, however, do not show this. Meanwhile several mechanisms of programmed cell death have been described in addition to apoptosis and necrosis. The inconclusive results of this study may be explained by another mechanism of programmed cell death?
Review of such a carelessly written, sloppy manuscript is an imposition and has been an annoying excercise! The text is full of typographical errors and written in a bad English style. I recommend to use the English spell check option of your word processing program. Part of the manuscript – especially the discussion section – is hardly comprehensible. The bad English style impairs the readability.
The arrangement of the figure panels in Figures 4-9 (i.e. morphological and nucleolar changes, PI staining HeLa, TUNEL HeLa, Western blot, PI staining GMK, TUNEL GMK, each figure with all 4 virus variants) does not match the order of description of the results in the text. Hence, reading was extremely tedious as I had to page up and down all the time.
Specific coments:
line 19: "deletion of either protein". Deletion is an inappropriate term here. You can delete a gene not a protein.
line 20 and elsewhere in the text: "elucidated" is an inappropriate term
line 22: typo observed
line 32 and line 518/519: print Cardiovirus and Picornaviridae in italics
line 33: typo picornaviruses
line 33: typo its
line 42: it should read "picornavirus infections"
line 63: it should read "cleaved"
lines 96/97: did the authors mean "indeed"?
line 78ff: L protein consists ot 67 amino acids, how many amino acids comprise the 2A protein?
line 212: explain HRP
lines 235-237, alternatively line 248: Describe the plaque phenotype of the mutant viruses.
line 247: Plato is a Greek philosopher. Did you mean plateau?
line 252: typo simultaneous
Figures 2, 3: Figure 2 presents data of BHK cells in panel A and HeLa cell results in panel B. In contrast, Figure 3 presents HeLa data in panel A and BHK data in panel C. It would be a good idea to present in both figures data of the cell lines in the same order.
line 312: it should read "underwent"
line 322: typo "Mock – uninfected cells"
line 324: it should read "intected"
line 334: it should read "nonfunctional"
line 344: typo characteristics
line 350-351: I can hardly detect green nuclei in Zfmut-infected BHK cells in Figure 9. Very weak staining!
line 353: type functional
line 366: bad English style: "our experiments can more fully reveal..."
line 368: typo expectations
line 393: typo analysis
line 395: it should read "Processed caspase-3 bands are marked..."
line 395: typo abnormal
line 400: "combined signs of both apoptotic and necrotic cell death" Is this possible? Did you check other forms of programmed cell death like pyroptosis, necroptosis, nemosis etc.?
line 446: typo amelioration
line 449: typo apoptosis
line 454: typo respond
line 455-456: Rewording required. This sentence is hardly comprehensible.
line 458-459: Rewording required. It should read like "... at which cytopathogenic signs develop..."
line 459: typo suppression
line 477: typo simultaneous
line 477: typo further
line 480: typo simultaneous
line 482: what is prorogate?
line 484: typo security
line 485: typo additional
line 485: typo permeabilizing
line 485: typo envelope
Figure 11: The stars look like white dots. Use another symbol.
line 509: typo occurs
line 513: typo disrupting, better: disruption
line 514: typo solely
line 519: anti-antiviral tools. The authors may consider the term "proviral".
line 522: it should read "design of viruses"
line 535: did you mean "... that L and 2A proteins are not directly involved..."?
line 536: it should read "...remain still viable..."
line 537-539: bad style, rephrase sentence
line 541-544: bad style, rephrase sentence
line 546: defected? did you mean defective?
line 548: it should read "one of the common cellular responses, which activate..."
line 627: it should read "...does not depend on it."
Comments on the Quality of English LanguageReview of such a carelessly written, sloppy manuscript is an imposition and has been an annoying excercise! The text is full of typographical errors and written in a bad English style. I recommend to use the English spell check option of your word processing program. Part of the manuscript – especially the discussion section – is hardly comprehensible. The bad English style impairs the readability.